# Dietary *Scutellaria baicalensis* and *Lonicera japonica* Extract Supplementation Attenuates Oxidative Stress and Improves Reproductive Performance in Sows

**DOI:** 10.3390/ani15243517

**Published:** 2025-12-05

**Authors:** Nuan Wang, Huiyuan Lv, Wei Chai, Hanting Ding, Junjie Yang, Hanyu Jing, Fang Chen, Wutai Guan

**Affiliations:** 1Guangdong Province Key Laboratory of Animal Nutrition Control, College of Animal Science, South China Agricultural University, Guangzhou 510642, China; lucaswong6@foxmail.com (N.W.); chaiwei98@163.com (W.C.); dinghanting@stu.scau.edu.cn (H.D.); a2625644309@163.com (J.Y.); 13833615379@163.com (H.J.); chenfang1111@scau.edu.cn (F.C.); 2State Key Laboratory of Animal Nutrition, College of Animal Science & Technology, China Agricultural University, Haidian District, Beijing 100193, China; lvhuiyuan0507@163.com; 3College of Animal Science and National Engineering Research Center for Breeding Swine Industry, South China Agricultural University, Guangzhou 510642, China

**Keywords:** sow, *Scutellaria baicalensis* and *Lonicera japonica*, oxidative stress, immune function, inflammatory regulation

## Abstract

In the late stages of pregnancy and during lactation, sows experience a marked increase in metabolic activity to support fetal growth and milk production. This heightened metabolic demand raises energy requirements and promotes catabolic processes, often leading to oxidative stress, weakened immune function, and negative impacts on reproductive performance as well as piglet growth. To address these challenges, this study evaluated the use of *Scutellaria baicalensis* and *Lonicera japonica* (SL) extracts as dietary supplements for sows. The results showed that chlorogenic acid and baicalin, which are key bioactive components in SL, effectively reduced endotoxin levels, lowered inflammatory markers, enhanced antioxidant capacity, and ultimately improved sow reproductive performance and offspring growth.

## 1. Introduction

In late pregnancy and lactation, sows face three main challenges: increased nutrient and energy needs for fetal growth and milk production [1,2], heightened maternal metabolism, which can cause oxidative stress, and metabolic fluctuations. These issues compromise immune function and raise risks such as porcine reproductive and respiratory syndrome virus (PRRSV) and miscarriage [3]. These challenges are exacerbated by climate, season, and poor management conditions. Impaired sow health during lactation can alter milk composition, impacting the piglets’ immunity, health, and growth [4,5].

Chinese herbal medicine, particularly *Lonicera japonica* from Southern China, has become popular as a feed additive for its minimal side effects and numerous benefits. It contains over 2.0% chlorogenic acid, a strong antioxidant that combats oxygen free radicals and lipid peroxidation [5,6]. Chlorogenic acid also shields cellular integrity from oxidative harm by eliminating hydrogen free radicals [7]. There is empirical proof of *Lonicera japonica*’s efficacy in livestock: Zhang et al. [8] revealed that its extract could significantly reduce oxidative stress markers in pigs and raise antioxidant enzyme activity after they had contracted PRRSV. Similarly, chlorogenic acid was shown to alleviate oxidative stress and inflammation in lipopolysaccharide (LPS)-stimulated bovine mammary epithelial cells, indicating therapeutic promise for treating mastitis [8]. Beyond merely its antioxidant attributes, *Lonicera japonica* also hosts antiviral, antibacterial, and anti-inflammatory effects. It suppresses inflammatory cytokines, including Tumor necrosis factor α (TNF-α) and Interleukin 18 (IL-18), by inhibiting the activation of the NF-κB and MAPK pathways so as to upgrade immune regulation [9,10]. *Scutellaria baicalensis*, another traditional Chinese medicine, contains baicalin (over 9.0%), a flavonoid known for its anti-inflammatory and antioxidant effects [11]. Baicalin enhances antioxidant enzymes like Superoxide Dismutase (SOD) and Catalase (CAT), inhibits oxidases, and neutralizes oxidative substances [12]. In broilers, baicalin could directly block reactive oxygen species (ROS) generation and reduce lipid peroxidation [13]. It also slows pathogenic bacteria and decreases pro-inflammatory cytokines, such as IL-1β, 6, and 8 [14]. The combined extracts may offer synergistic benefits. Zhao et al. [15] found that SL extract increased Total Antioxidant Capacity (T-AOC) levels and reduced malondialdehyde (MDA) content in weaned piglets, indicating a joint enhancing effect. Furthermore, Zhang et al. [16] and Li et al. [17] confirmed the safety and efficacy of these herbs in broilers and weaning piglets. They suggested that the observed improved growth performance may have been due to the regulation of gut microbiota, which improves the digestibility of nutrients.

Several studies have confirmed the safety and efficacy of these herbs in different animals and situations. However, existing studies have not clarified how SL additives enhance piglet performance by improving sow health and milk quality during late pregnancy and lactation. The transmission of immunity, antioxidant, and anti-inflammatory capacities from sow to offspring has recently been a major research focus.

Consequently, this study was conducted from day 85 of gestation in sows through to day 21 of lactation, with the objective of examining the effects of SL additives on sow health, reproductive performance, milk composition, endocrine hormones, and the innate immunity of the offspring. In addition, in vitro experiments were performed on porcine mammary epithelial cells (pMECs). The concentration of SL additives was meticulously maintained at a consistent level across both in vitro and in vivo experiments to facilitate a comprehensive investigation of its mechanisms of action.

## 2. Materials and Methods

### 2.1. Materials

The extract powder of SL was combined with 2 major components, with the content of chlorogenic acid effective substances being ≥2.2‰, and the content of baicalin effective substances being ≥2.2%. Cell experiment: chlorogenic acid and baicalin were bought from Sigma (St. Louis and Burlington, MA, USA), 94419 and 572667, and the mixing ratio was 1:10.

Dulbecco’s Modified Eagle Medium: Nutrient Mixture F-12 (DMEM/F12, Gibco, Grand Island, NY, USA); fetal bovine serum (Gibco, USA); PBS solution (Gibco, USA); penicillin-streptomycin (Sigma, USA); 0.25% trypsin (Sigma, USA); hydrocortisone (Sigma, USA); IGF-1 (Sigma, USA); insulin-transferrin-selenium (ITS, ScienCell, Carlsbad, CA, USA); EGF (PeproTech, Cranbury, NJ, USA); CCK-8 (Nanjing, China, Jiancheng); pure LPS (Escherichia coli 055:B5, Sigma, USA); pure neochlorogenic acid (>98%, Sigma, USA); pure baicalin (>95%, Sigma, USA).

### 2.2. Methods

#### 2.2.1. Animals and Experimental Design

The feeding experiment took place at the Longkou pig farm, Guangdong Daguang Farming Co., Ltd. (Jiangmen, Guangdong, China), from 1 July to 16 September 2021. Healthy, multiparous sows of the “Duroc × Landrace × Large White” breed with three to six parities were selected at day 85 of pregnancy (n = 100). They were evenly divided into two groups based on delivery date, fatness, parity, and past reproductive performance. Each group had 50 replicates with one sow each. The experiment had a single-factor design: (1) Control group with a basal diet; (2) Experimental group with a basal diet plus 0.05% SL extract (Table 1).

#### 2.2.2. Diet and Feeding Management

The basal diet for this trial was a corn–soybean meal formulation based on NRC 2012 [18] nutrient standards (Table 2). Sows were allotted in gestation pens (2.15 × 1.15 m) and fed twice daily at 06:30 and 14:30, with ad libitum water. On day 110 of pregnancy, they were moved to the farrowing house, where limited feeding occurred at 1.0 kg (day of farrowing) plus 0.5 kg/day. Limited feeding continued until day 6 of lactation. From days 6 to 21 of lactation, sows were fed ad libitum, and their feed intake was closely monitored. Sows were cross-fostered within groups 24 h post-farrowing, with 10 to 12 pigs per group.

#### 2.2.3. Data Recording and Sample Collection

When sows farrow, the total piglets born, including stillbirths, mummies, live, healthy, and weak piglets (≤0.8 kg), along with litter and individual birth weights, were recorded. On the 21st day, piglet weaning weights, sow estrus 7 days post-weaning, and piglet average daily weight gain during lactation were noted. Sow feed consumption during the 21-day lactation was tracked to calculate average daily feed intake (ADFI). Backfat thickness was measured at the P2 point (6 cm from the midline of the 10th rib on the left side), using an ultrasonic device (Renco Lean-Meater^®^, Renco Corporation, Minneapolis, MN, USA), on the 1st and 21st days of farrowing, to determine average backfat loss during lactation.

On gestation day 85, farrowing day, and days 14 and 21 post-farrowing, 10 healthy sows from each group, representing average production performance, were selected. About 10 mL of blood was drawn from their ear vein, placed in EDTA tubes (Becton Dickinson, Franklin Lakes, NJ, USA), and centrifuged at 25 °C at 3500× *g* for 15 min. The plasma was transferred to sterile cryotubes (Becton Dickinson, Franklin Lakes, NJ, USA), snap frozen in liquid nitrogen, and stored at −80 °C for further research. Similarly, on farrowing day and weaning day 21, 6 average piglets from each group had blood collected from the anterior vena cava, processed, and stored using the same method as the sows.

#### 2.2.4. Measurement of Antioxidant, Inflammatory, and Immunoglobulin Indicators

Concentrations of IL-1β (CSB-E06782p), IL-6 (CSB-E06786p), IL-8 (CSB-E06779p), IL-18 (CSB-E13864p), T-AOC (A015-1), MDA (A003-2), SOD (A001-3), GSH-Px (A031-3), and GSH (A030-1) in plasma were measured with the ELISA kit purchased from Nanjing Jiancheng Bioengineering Institute (Nanjing, China) following the instructions of the manufacturers. IgA (CSB-E13234p), IgG (CSB-E12734G), and IgM (CSB-E12935f) content was assessed per Wuhan Huamei (Wuhan, China) ELISA kit guidelines. Milk samples were taken within 12 h post-farrowing and on the 14th day of lactation. For the latter, 15 IU oxytocin was injected into the ear vein before collection. About 30 mL of milk was collected each time, transferred to 15 mL sterile tubes, and stored in liquid nitrogen before being moved to a −80 °C freezer. Whey was extracted by centrifuging the milk at 3000× *g* for 20 min at 4 °C, following Zanello et al.’s [19] method.

### 2.3. In Vitro Cell Experiments

#### 2.3.1. Cell Recovery and Passage

Remove 1 mL of frozen pMECs from liquid nitrogen using tweezers and thaw them in 37 °C water for 3 min. Transfer the thawed cells to a sterile tube with 9 mL of culture medium. Mix, then centrifuge at 1000 r/min for 3 min. Discard the supernatant, rinse the cells with 9 mL PBS, and centrifuge again. Repeat the rinse and centrifuge process once more. Finally, add 10 mL of culture medium, mix to achieve a uniform cell concentration, and adjust the cell density to 1 × 10^5^/mL. Inoculate into a 25 cm^2^ sterile culture bottle, shake gently, and place in a 37 °C incubator. Once cell coverage reaches 80%, subculture onto a 12-well plate.

#### 2.3.2. Inflammation Model

We used the Cell Counting Kit-8 (CCK-8, Sigma, USA) method to evaluate how different concentrations and exposure times of bacterial lipopolysaccharide (a) affect pMEC viability. Once pMECs were about 80% confluent, they were digested with 0.25% trypsin, centrifuged, and counted. The cell suspension was adjusted to 2 × 10^4^ cells/mL and inoculated into a 96-well plate with 200 μL per well, then incubated at 37 °C with 5% CO_2_ for 48 h. After removing the medium, fresh medium with varying LPS concentrations (0, 1, 5, 10, 25, 50, 100, 200 μg/mL) was added, creating 8 groups with 6 replicates each. The plate was incubated again for 12, 24, and 48 h. Absorbance at 450 nm was measured using a CCK-8 kit and a microplate reader (Thermo Scientific, Wilmington, NC, USA) to calculate cell viability. Apoptosis was tested using a flow cytometer (Mindray, Shenzhen, China). Cell viability, apoptosis, and inflammatory factor expression were evaluated to establish the inflammatory model conditions.

#### 2.3.3. Effects of Different Concentrations of SL Extract on pMEC Viability

The CCK-8 method assessed the impact of varying *Scutellaria baicalensis* extract concentrations on cell viability. When pMEC reached 70–80% confluence, cells were digested with 0.25% trypsin, centrifuged, and counted. The cell suspension was adjusted to 1 × 10^4^ cells/mL, and 200 μL was added per well in 96-well plates and incubated at 37 °C with 5% CO_2_ for 48 h. Different extract concentrations (0–100 μg/mL) were then added, with six replicates per group, and incubated for another 24 h. Afterward, 20 μL of CCK-8 reagent was added to each well, and incubation continued for 4 h. Absorbance at 450 nm was measured using an automatic microplate reader (Thermo Scientific, USA).

### 2.4. Protein Blotting

For electrophoresis, use a 10% or 12% sodium dodecyl sulfate–polyacrylamide gel electrophoresis (SDS-PAGE) gel (P0012AC, Beyotime, Shanghai, China) to separate proteins based on their molecular weight. Transfer the target proteins to a polyvinylidene difluoride (PVDF) membrane using a tri-glycine system. Rinse the membrane three times with tris-buffered saline with 0.1% Tween^®^ 20 detergent (TBST) buffer, then block with 5–6% skim milk at 26 °C for 2 h. After blocking, rinse again with TBST buffer, apply the primary antibody, and incubate overnight at 4 °C. Rinse the membrane three times, add the secondary antibody, and incubate at 26 °C for 1 h. Finally, use super electrogenerated chemiluminescence (ECL) reagent (P1020 ApplyGen, Beijing, China) to detect chemiluminescent signals and quantify them with ImageJ software (LOCI, University of Wisconsin, WI, USA, version 1.52a).

### 2.5. Real-Time Fluorescence Quantitative PCR

RNA was extracted from the cell line using the EZB kit (EZbioscience, Roseville, MN, USA). The concentration and quality of the extracted RNA were determined using a NanoDrop ND-1000 spectrophotometer (Thermo Scientific, Wilmington, NC, USA). The RNA was then reverse transcribed into cDNA using a reverse transcriptase kit (A0010CGQ, EZBioscience, Roseville, MN, USA). RT-PCR was performed to assess inflammatory factor mRNA expression, using β-actin as a control. The PCR cycle was as follows: 95 °C for 30 s, followed by 40 cycles of 95 °C for 15 s and 60 °C for 30 s. The primer design for this experiment was synthesized by Sangon Biotech Co., Ltd. (Shanghai, China), conducted in accordance with the methodology outlined in a previously published article by Zhang et al. [20] and Tian et al. [21] from our research group. Results were analyzed with the 2^−ΔΔCt^ method [22], and primer details are in Table 3.

### 2.6. Statistical Analysis

Reproductive performance was analyzed with SPSS 23.0 (IBM, Armonk, New York, NY, USA), presented as mean and standard error. An independent *t*-test was used to analyze the differences between the two different treatments. The residue normality of the variance of the in vitro cell studies was determined by the Shapiro–Wilk test, then analyzed by one-way ANOVA. Duncan’s method was used to make multiple comparisons. Tendencies were noted at *p* < 0.10, significant differences at *p* < 0.05, and highly significant differences at *p* < 0.01. Graphs were drawn and analyzed using the GraphPad Prism 8.0 software (La Jolla, CA, USA).

## 3. Results

### 3.1. Reproductive and Farrowing Performance

In comparison to the control group, the inclusion of SL extract powder in the diet resulted in an 8.70% increase in the litter weight at birth, achieving statistical significance (*p* < 0.05, Table 4). However, no significant differences were observed in the total number of piglets born, the number of live piglets, the number of healthy piglets, the number of weak piglets, the incidence of stillbirths, the number of mummified fetuses, or the individual weight of the first live piglets (*p* > 0.05, Table 1).

### 3.2. Milk Composition and Lactation Performance

Adding SL extract powder to the diet did not significantly affect colostrum and 14th-day milk composition in lactating sows compared to the control group (*p* > 0.05, Table 5). However, it significantly improved the survival rate of weaned piglets (*p* < 0.05, Table 6) and greatly increased sows’ average daily feed intake during lactation (*p* < 0.01, Table 6). There were no significant changes in litter weight, individual piglet weight, number of weaned piglets, piglet weight gain, sows’ back fat loss during 21 days of lactation, or estrus interval (*p* > 0.05, Table 7).

### 3.3. Plasma Hormone Levels

Compared with the control group, the incorporation of SL extract powder into the sows’ diet significantly elevated the plasma prolactin levels in the sows on the day of farrowing and on the 14th day of lactation (*p* < 0.10, Table 8), with increases of 8.15% and 5.98%, respectively. However, no significant differences were observed in the levels of corticosterone, insulin, glucagon, T3, T4, and leptin (*p* > 0.05, Table 8). Furthermore, the addition of SL extract powder was associated with a reduction in plasma endotoxin levels in the sows on the 14th and 21st days of lactation (*p* < 0.10, Table 9), with decreases of 4.40% and 5.69%, respectively. Additionally, a significant reduction in plasma HSP-70 levels was noted on the 14th day of lactation (*p* < 0.10, Table 9), with a decrease of 7.57%.

### 3.4. Antioxidant Function of Sows and Piglets

According to Table 10, Table 11 and Table 12, compared with the Control group, the inclusion of SL extract powder in the diet of the lactating sows resulted in a statistically significant enhancement of total antioxidant capacity in the plasma, colostrum, milk on day 14, and plasma on day 21 of lactation (*p* < 0.05), with increases of 13.21%, 9.09%, 5.99%, 12.12%, and 18.52%, respectively. Additionally, there was an observed increase in total antioxidant capacity in the plasma of the sows on the day of farrowing and in the plasma of the newborn piglets (*p* < 0.10), with increases of 10.91% and 26.0%, respectively. Furthermore, the supplementation significantly reduced malondialdehyde (MDA) content in the plasma, colostrum, and plasma of the newborn piglets on the day of farrowing, as well as on the 14th and 21st days of lactation (*p* < 0.05), with reductions of 9.97%, 10.03%, 12.38%, 23.01%, and 8.88%, respectively. It also decreased MDA content in the normal milk of the sows on the 14th day and in the plasma of the piglets on the 21st day (*p* < 0.10), with reductions of 16.02% and 12.76%, respectively.

### 3.5. Immune Function of Sows

In comparison to the control group, the inclusion of SL extract powder in the diet of the sows resulted in an increase in plasma levels of Ig (immunoglobulin)A, IgG, and IgM by 9.59%, 8.93%, and 9.21%, respectively, on the 14th day of lactation (*p* < 0.10, Table 13). Additionally, the IgM content in milk on the 14th day showed an increase of 12.07% (*p* < 0.10, Table 14). However, there was no significant effect on the levels of IgA and IgG in colostrum and milk on the 14th day (*p* > 0.05).

### 3.6. Inflammatory Factor Content in Sows

There was a significant decrease in the levels of TNF-α (L0), IL-1β (L14), IL-6 (L0), and IL-8 (L21) in sow plasma, with reductions of 5.71%, 8.29%, 8.15%, and 6.97%, respectively (*p* < 0.05, Table 15). Additionally, the incorporation of SL extract powder into the diet of sows significantly reduce the IL-8 concentration in their milk by 9.46% on the 14th day (*p* < 0.01, Table 16). Furthermore, a downward trend was observed in the concentrations of TNF-α (L14), IL-6 (L14), and IL-18 (L0/L14) in sow plasma, as well as TNF-α and IL-18 in their colostrum, with respective decreases of 8.12%, 6.11%, 8.22%, 6.90%, 6.56%, and 9.11% (*p* < 0.10, Table 16). The supplementation also significantly increased the soluble cluster of differentiation 14 (sCD14) concentration in sow plasma, by 10.29% on the 14th day (*p* < 0.05, Table 15), and showed an upward trend in sCD14 levels in sow colostrum, with an increase of 10.24% (*p* < 0.10, Table 16). Similar tendency was shown in piglets, as the levels of TNF-α (L0) decreased by 10.70% (Table 17).

### 3.7. Establishment of In Vitro Inflammatory Model

According to Figure 1A, the cell viability of pMECs exhibited a general decline with increasing concentrations of LPS and prolonged treatment duration. An optimal LPS concentration of 50 μg/mL and a treatment duration of 24 h were identified for further investigation. Under these conditions, a significant increase in cell apoptosis was observed (*p* < 0.05, Figure 1B). Concurrently, the expression levels and content of inflammatory factor mRNA were assessed, revealing a significant upregulation at this time point (*p* < 0.05, Figure 1C,D). Considering the parameters of cell viability, apoptosis, and inflammatory factor expression, the conditions selected for establishing the inflammatory model were an LPS concentration of 50 μg/mL and a treatment duration of 24 h.

### 3.8. Effect of SL Extract on pMEC Viability

The effects of SL extract on pMEC viability are shown in Figure 2. The viability of pMECs first increased and then decreased in the range of 0–5 μg/mL. Cell viability was highest at 2 μg/mL. Therefore, 2 μg/mL was selected as the optimal pretreatment concentration of SL extract in this experiment. Pretreatment of pMECs with 2 μg/mL SL extract for 24 h significantly alleviated the decrease in cell viability induced by 50 μg/mL of LPS (*p* < 0.05, Figure 3).

### 3.9. Effects of SL Extract on Inflammatory Factors, Antioxidant Status, and Inflammatory Signaling Pathways in pMECs Induced by LPS

A 24 h pretreatment with SL extract significantly inhibited the LPS-induced upregulation of both mRNA expression and the content of inflammatory factors of TNF-α, IL-1β, IL-6, and IL-8, thereby alleviating cellular inflammatory damage (*p* < 0.05, Figure 3). Moreover, pretreatment with SL extract resulted in a notable increase in total antioxidant capacity (T-AOC) and superoxide dismutase (SOD) levels in cells subjected to inflammatory damage, while simultaneously decreasing intracellular malondialdehyde (MDA) content (*p* < 0.05, Figure 4).

LPS exposure led to a significant elevation in the phosphorylation levels of proteins involved in the NF-κB and MAPK inflammatory signaling pathways (*p* < 0.05, Figure 4 and Figure 5). Nonetheless, SL extract pretreatment effectively inhibited the phosphorylation of transcription factor p65 (p65), inhibitor kappa B alpha (IκBα), p38 mitogen-activated protein kinase (p38), extracellular signal-regulated kinase (ERK), and Jun N-terminal kinase (JNK) proteins, thereby substantially mitigating the inflammatory response in mammary epithelial cells (*p* < 0.05, Figure 4 and Figure 5).

## 4. Discussion

### 4.1. Challenges for Sows in Late Gestation and Lactation

Animal health relies on sustaining a balanced physiological state. Sows face metabolic changes that can disrupt homeostasis from late pregnancy to lactation. At this time, they are highly vulnerable to oxidative stress, immunosuppression, and inflammation. These issues can be further exacerbated by high temperature and humidity in the summer, intensifying the influence of these negative factors. A key hormone for fetal growth and milk production, prolactin, was first identified in the cow and subsequently in humans. It is produced through the H-P-A axis and regulated by the nervous and hormonal systems [23]. If conditions of oxidative stress persist, the subsequent reduction in prolactin levels usually results in negative outcomes, including smaller litter size, lower survival rate of piglets, and decreased vitality among the piglets [24,25]. In the presence of such virulent diseases as PRRSV and Porcine Epidemic Diarrhea (PED), as the health of sows in late pregnancy is threatened, the number of total-born, live-born, and healthy piglets decreases significantly [26,27].

### 4.2. Antioxidant Effects of SL Extract

Chlorogenic acid and baicalin in SL extract possess potent antioxidant activity, reducing deleterious free radicals and protecting the body from oxidative stress [28]. Baicalin, the main active compound of *Scutellaria baicalensis,* also exhibits powerful antioxidant activity, enhancing antioxidant enzymes and decreasing the activity of oxidase [29]. It has been found that chlorogenic acid (600 mg/kg) can improve T-AOC and T-SOD levels, reduce MDA levels in heat-stressed chicks, and alleviate inflammatory bacteria colonization [30]. In weaned piglets, 10,000 mg per kg chlorogenic acid supplementation increases GSH-Px and CAT, while decreasing MDA. It controls the impairment of the intestinal barrier caused by weaning stress [31].

### 4.3. Anti-Inflammatory and Immunomodulatory Effects of SL Extract

Other signs of elevated oxidative stress burden include increasing inflammatory cytokines and a decrease in immune function. Levels of immunoglobulins reflect the immune capacity [32]. Studies have shown that SL extract added to the feed of broilers can raise the content of serum IgA and IgG, enhance their immunity, and improve production performance [33]. Dietary inclusion of 0.5 g/kg chlorogenic acid in the feed reduces serum IL-6 and IL-10 levels and improves IgA concentration in broilers, subsequently producing positive effects in their immunity and growth performance [34]. Furthermore, baicalin decreases the level of serum IgE and histamine in mice with nasal sinusitis, thus enhancing their immune response, and has a therapeutic effect [35]. Supplementation with baicalin in the diet reduced the serum concentrations of IgE, IL-1β, IL-4, IL-6, TNF-α, and histamine in rats with allergic rhinitis, which suggests its potential as a therapeutic agent for allergic rhinitis through inhibition of inflammation mediators [36]. Chlorogenic acid attenuates Dextran Sulfate Sodium Salt (DSS)-induced ulcerative colitis in mice by decreasing inflammation and cell apoptosis in the colon through inhibition of the MAPK/ERK/JNK signaling pathway [37]. It reduces levels of TNF-α, IL-1β, and mRNA expressions of these cytokines in hyperuricemic mice [38].

### 4.4. Intergenerational Transmission of Health Inequalities

If the late pregnancy health of sows is impaired, the proportion of total, live, and healthy litters is decreased compared to that of normal sows [39]. Furthermore, the feed intake and milk composition of sows are altered, which causes negative effects on the weaning weight of piglets when challenged by stress during lactation [40]. Additional experimental data show that weak piglets with lower birth weight probably suffer further developmental retardation and fail to keep up with the growth of their healthier littermates within the same litter [41]. Therefore, we hypothesized that these susceptible piglets nursed by sows subjected to stress that produces low-quality milk will exhibit even greater decreases in growth and total output potential.

In the present study, results indicated that sows and piglets had markedly higher antioxidant levels when the sows’ ration was supplemented with SL extract during late gestation and lactation, and their MDA content was decreased by 33.29% compared to the control group. These findings indicate that SL extract has a potent antioxidant and anti-inflammatory activity. Also, the addition of SL had a minor influence on immunoglobulin levels in sow plasma, milk, and piglet plasma. Additionally, it significantly increased IgG, IgA, and IgM concentrations in sow plasma on day 14 of lactation (*p* < 0.05), which is consistent with other results. Also, TNF-α, IL-1β, and IL-6 in sow plasma and IL-8 in normal milk and IL-8 and IL-18 in sow plasma, colostrum, and piglet plasma were lower when supplemented by dietary SL extract. The reduction in pro-inflammatory factors and increase in immunoglobulins indicate that SL extract has strong anti-inflammatory properties. This effect supports its potential for clinical use. The enhancement of immune functions is possibly attributed to the antioxidant, anti-inflammatory, and immunoregulatory functions of SL extract. Additionally, the present study revealed that adding 0.5 kg/T of SL extract to the sows’ diet could increase the litter weight of newborn piglets (*p* < 0.10), significantly enhance the survival rate of weaned piglets (*p* < 0.05), and dramatically increase the sows’ feed intake during lactation (*p* < 0.01).

### 4.5. Antioxidant and Immunoregulatory Effects on Porcine Mammary Epithelial Cells (PMECs)

In order to clarify the impact of sows’ antioxidant capacity and immune function on their milk and offspring, in vitro studies on porcine mammary epithelial cells (PMECs) were conducted.

Previous study shows that chlorogenic acid decreases TNF-α, IL-1β, and IL-6 in LPS-stimulated RAW 264.7 macrophages [42]. SL reduces levels of Nitric Oxide (NO), Prostaglandin E2 (PGE2), IL-12, IL-2, IL-6, IL-7, and TNF-α in LPS-induced Raw 264.7 cells and lowers inflammation and stimulates cell growth [43]. Baicalin attenuates LPS-induced microglial inflammation via suppressing NF-κB and MAPKs pathways’ phosphorylation in BV-2 microglia and plays a role by downregulating inflammation [44]. It decreases apoptosis and downregulates apoptosis-related genes in *Glaesserella parasuis* (GPS)-stimulated porcine peritoneal mesothelial cells (PMCs), alleviating GPS-induced inflammation by suppressing MAPK-pathway activation [45]. It has been shown that chlorogenic acid could downregulate phosphorylation of proteins of the MAPK and NF-κB pathways in Human gingival fibroblasts 1(HGF-1) cells, ameliorating LPS-induced inflammatory response [46]. Moreover, chlorogenic acid inhibits pro-inflammatory cytokines and mediators such as TNF-α, IL-6, IL-1β, Cyclooxygenase-2 (COX-2), inducible nitric oxide synthase (iNOS), and NO, and inactivates the NF-κB and MAPK pathway, suppressing cellular inflammatory injury [47]. The antioxidant activity of cells can be strengthened by increasing the activity of superoxide dismutase (SOD) and the production of nitric oxide (NO) via the nuclear factor erythroid 2-related factor 2 (Nrf2) pathway, thereby reducing the damage of oxidative stress [48]. Chlorogenic acid also inhibits the production of Reactive oxygen species (ROS) in IL-1β-treated human umbilical vein endothelial cells (HUVECs) and upregulates endogenous antioxidant capacity and partially preserves the activity of LPS-reduced autophagy in human chondrocytes by inducing autophagy and reducing ROS-induced cell apoptosis [49]. However, there are limited reports on the impact of SL extract on the anti-oxidative capacity of PMECs.

In this study, SL extract significantly raised T-AOC and SOD levels and lowered MDA (*p* < 0.05), thereby enhancing cellular antioxidant capacity and alleviating inflammation in sow mammary epithelial cells. Also, it had a significantly inhibitory effect on the expression of TLR4, p-P38, p-IκBα, p-p65, p-ERK, p-JNK, MAPK, and NF-κB pathways’ phosphorylation, which led to a dramatic reduction in inflammatory cytokines. These results are not only congruent with previous studies in various cells, but also with the results of our in vivo studies.

## 5. Conclusions

This experiment examined the effect of dietary SL extract in in vivo and in vitro models. It proved that adding SL extract powder to the diet of sows from late pregnancy to lactation improved their production performance by improving their antioxidant capacity and immunity and reducing the inflammation level of the sows. Part of the positive effect was transmitted to piglets through colostrum and milk, and, furthermore, enhanced the piglets’ performance.

## Figures and Tables

**Figure 1 animals-15-03517-f001:**
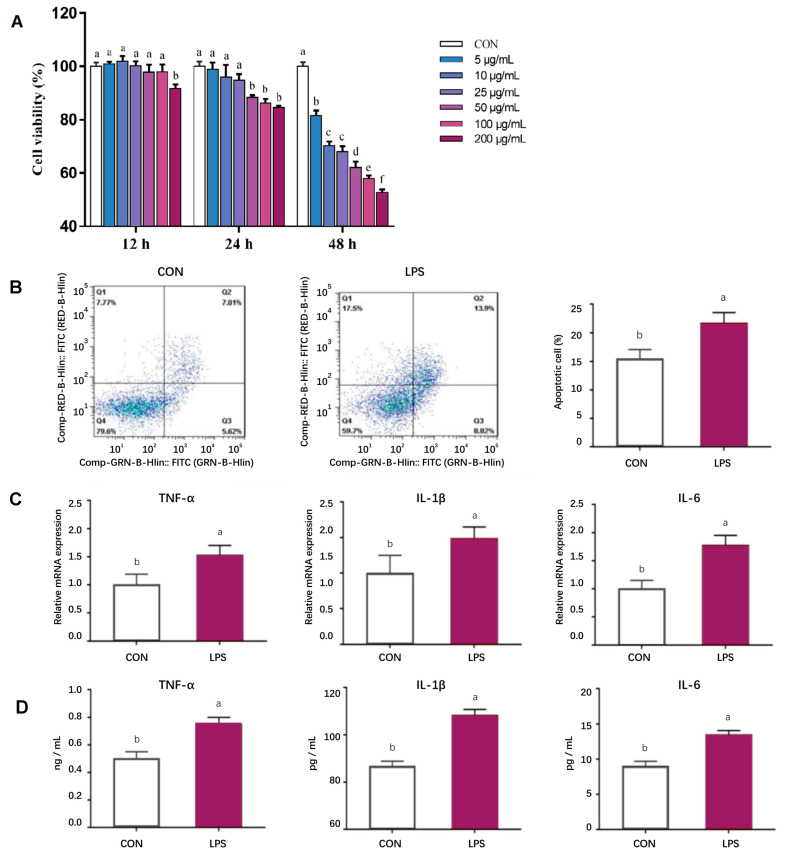
Establishment of in vitro inflammatory model. (**A**) Effects of LPS treatment at different levels (0, 5, 10, 25, 50, 100, 200 μg/mL) and times (12, 24, 48 h) on pMEC viability; (**B**) Effects of adding 50 μg/mL of LPS treatment for 24 h on pMEC apoptosis; (**C**) Effects of adding 50 μg/mL of LPS treatment for 24 h on the mRNA expression abundance of inflammatory factors in pMECs; (**D**) Effects of adding 50 μg/mL of LPS treatment for 24 h on the content of inflammatory factors in pMECs-related cells. Data are expressed as mean ± standard error, n = 6; Different superscript letters indicate significant differences (*p* < 0.05). pMEC: Porcine mammary epithelial cells. CON: Control group. LPS: Challenged by lipopolysaccharide.

**Figure 2 animals-15-03517-f002:**
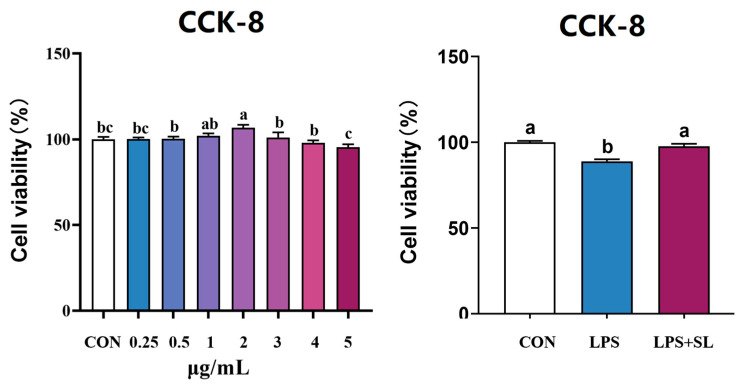
Effect of SL extract on pMEC viability. Data are expressed as mean ± standard error, n = 6; Different superscript letters indicate significant differences (*p* < 0.05). CCK-8: Cell Counting Kit-8. CON: Control group. LPS: Challenged by lipopolysaccharide. SL: *Scutellaria baicalensis* and *Lonicera japonica* treated.

**Figure 3 animals-15-03517-f003:**
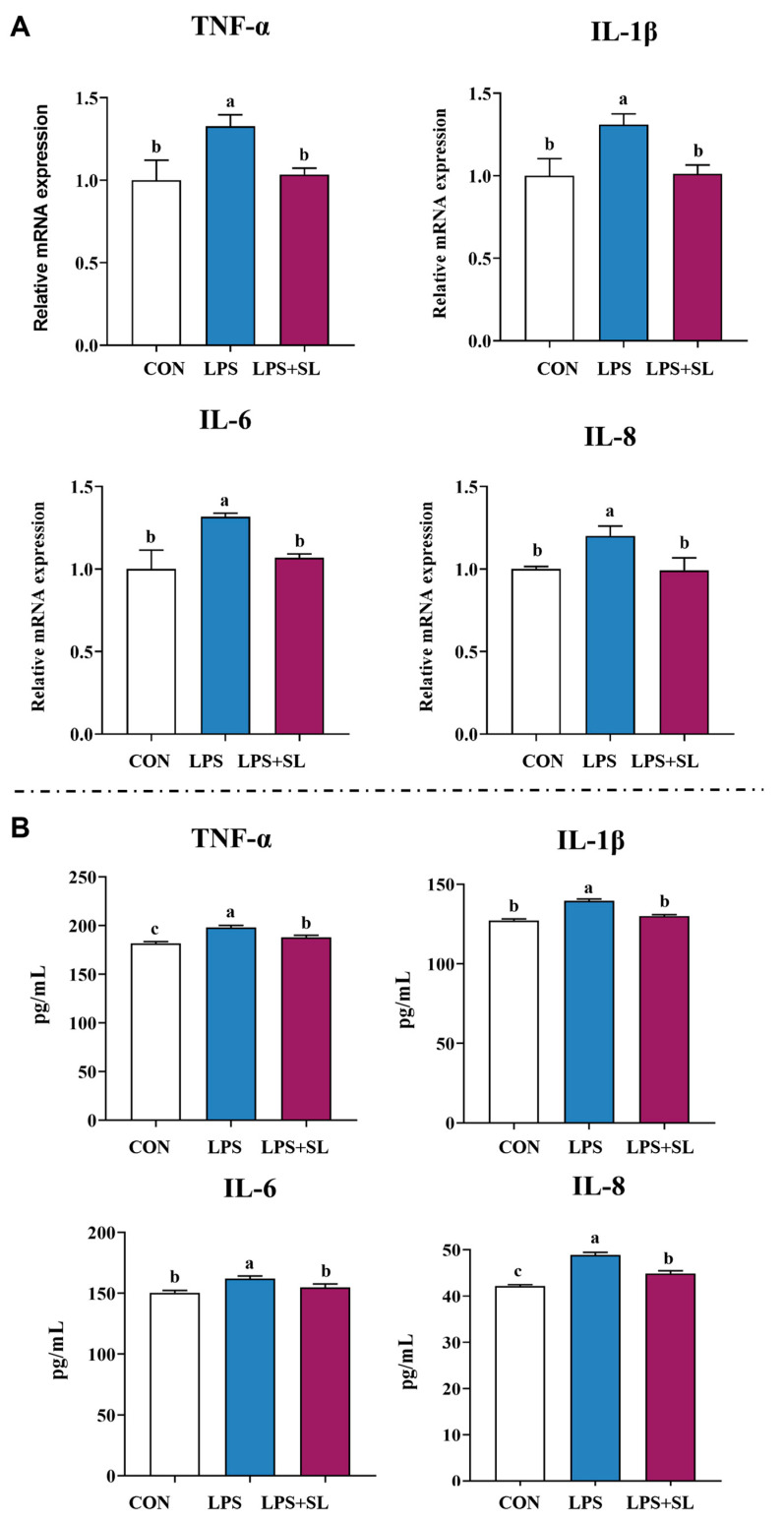
Effect of SL extract on relative mRNA expression (**A**) and content (**B**) of inflammatory cytokines. Data are expressed as mean ± standard error, n = 6; Different superscript letters indicate significant differences (*p* < 0.05). TNF-α: Tumor necrosis factor α. IL: Interleukin. CON: Control group. LPS: Challenged by lipopolysaccharide.

**Figure 4 animals-15-03517-f004:**
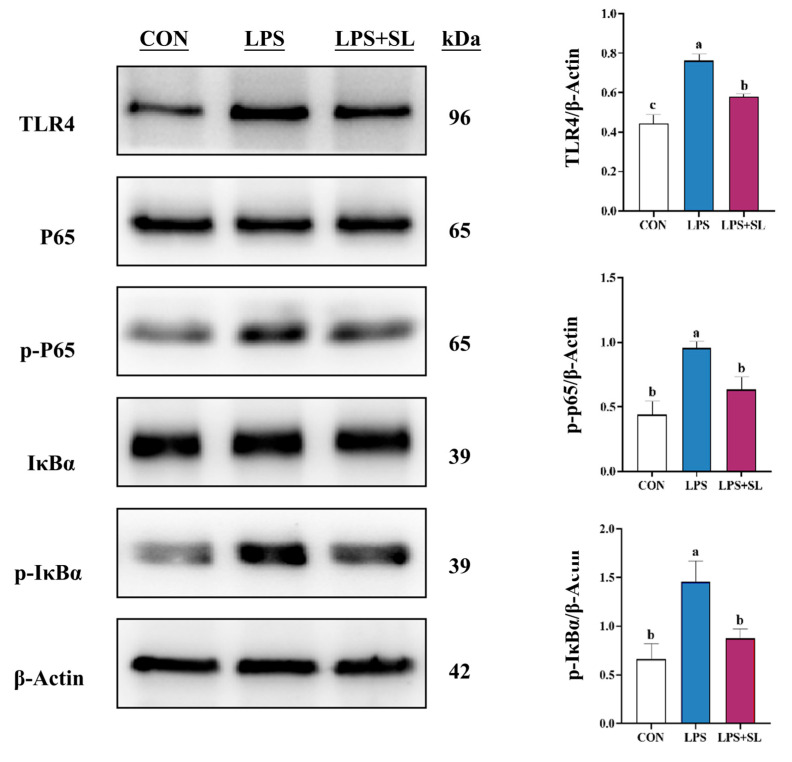
Effect of SL extract on NF-κB signaling pathway in pMECs. Data are expressed as mean ± standard error, n = 3; Different superscript letters indicate significant differences (*p* < 0.05). TLR4: Toll-like receptor 4. p65: Transcription factor p65. p-p65: Phosphorylated transcription factor p65. IκBα: Inhibitor kappa B alpha. p-IκBα: Phosphorylated inhibitor kappa B alpha. CON: Control group. LPS: Challenged by lipopolysaccharide. SL: *Scutellaria baicalensis* and *Lonicera japonica* treated. kDa: kilodalton.

**Figure 5 animals-15-03517-f005:**
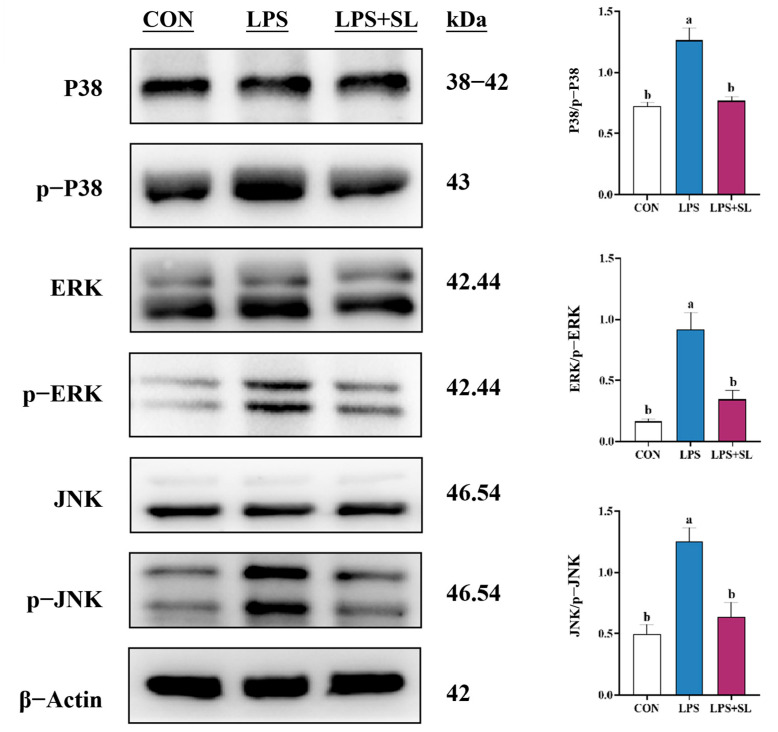
Effect of SL extract on MAPK signaling pathway in pMECs. Data are expressed as mean ± standard error, n = 3; different superscript letters indicate significant differences (*p* < 0.05). p38: p38 mitogen-activated protein kinase. p-p38: phosphorylated p38 mitogen-activated protein kinase. ERK: Extracellular signal-regulated kinase. p-ERK: Phosphorylated Extracellular signal-regulated kinase. JNK: Jun N-terminal kinase. p-JNK: phosphorylated Jun N-terminal kinase. CON: Control group. LPS: Challenged by lipopolysaccharide. SL: *Scutellaria baicalensis* and *Lonicera japonica* treated. kDa: kilodalton.

**Table 1 animals-15-03517-t001:** Division of the experimental group.

	Control Group	Trial Group (+0.05% SL Extract)
SL extract	0	+0.5 kg/t
Sows	50	50

SL: Scutellaria baicalensis and Lonicera japonica.

**Table 2 animals-15-03517-t002:** Diet composition and nutrient level (as fed basis)%.

Ingredients	Proportion (%)	Nutrient Levels	Content (%)
Corn	58.71	DE (MJ/kg)	14.31
Soybean Meal	24.00	Crude Protein	18.11
Wheat Middings	8.00	Crude Fiber	3.15
Fish meal	2.00	Crude Ash	5.92
Soybean Oil	4.00	Ether Extract	6.82
Ca(H_2_PO_4_)_2_	0.30	Calcium	1.02
Limestone	1.60	Total Phosphorus	0.79
Mineral Premix ^1^	0.10	Available Phosphorus	0.52
NaCl	0.30	Lysine	0.84
Na_2_SO_4_	0.40	Methionine + Cysteine	0.49
NaHCO_3_	0.20	Tryptophan	0.18
Vitamin Premix ^2^	0.04	Threonine	0.68
Vitamin C (95%)	0.02		
Biotin (2%)	0.005		
Folic acid (98%)	0.005		
Choline chloride (50%)	0.20		
L-Lysine (98–99%)	0.10		
Phytase	0.02		
Total	100.0		

(^1^) Mineral premix provided the following per kilogram of complete diet: 80 mg iron, 5 mg copper, 51 mg zinc, 20.5 mg manganese, 0.14 mg iodine, 0.15 mg selenium. (^2^) Vitamin premix provided the following per kilogram of complete diet: 13000 IU VA, 4000 IU VD3, 60 mg VE, 4 mg VK3, 4 mg VB1, 4 mg VB2, 10 mg VB6, 4.8 mg VB12, 0.034 mg niacin, 40 mg D-pantothenic acid, 20 mg folic acid, 0.16 mg D-biotin. DE: Digestible energy.

**Table 3 animals-15-03517-t003:** RT-PCR primers.

Gene	Primer Sequence (5′-3′)	Size (bp)
*TNF-α*	ACGGGCTTTACCTCATCTACTC GGCTCTTGATGGCAGACAGG	141
*IL-1β*	CCGAAGAGGGACATGGAGAA AGTTGGGGTACAGGGCAGAC	88
*IL-6*	TGGCTACTGCCTTCCCTACC CAGAGATTTTGCCGAGGATG	132
*IL-8*	AGGACCAGAGCCAGGAAGAGAC CACAGAGAGCTGCAGAAAGCAG	108
*β-Actin*	TGCGGGACATCAAGGAGAAG AGTTGAAGGTGGTCTCGTGG	176

*TNF-α*: Tumor necrosis factor α. *IL-1β*: Interleukin 1β. *IL-6*: Interleukin 6. *IL-8*: Interleukin 8.

**Table 4 animals-15-03517-t004:** Effects of dietary SL extract powder on reproductive performance of sows.

	CON	SL	SEM	*p*-Value
Replicates	49	48		
Average Parity	3.43	3.44	0.07	0.930
Total born	11.37	12.00	0.39	0.255
Born alive	9.33	10.06	0.35	0.141
Healthy Piglets ^1^	9.20	9.88	0.33	0.157
Weak Piglets ^2^	0.12	0.19	0.08	0.574
Stillborn	1.49	1.27	0.17	0.359
Mummified	0.55	0.67	0.14	0.553
Litter Weight, kg	14.95 ^b^	16.25 ^a^	0.53	0.047
Birth Weight, kg	1.62	1.62	0.03	0.900

(^1^) Piglets with birth weight > 0.8 kg. (^2^) Piglets with birth weight < 0.8 kg. One sow died in the control group during pregnancy, and two sows aborted in the experimental group. Different lowercase letters in superscripts in the same line indicate significant differences (*p* < 0.05), while no letters indicate no significant differences (*p* > 0.05). CON: Control group. SL: *Scutellaria baicalensis* and *Lonicera japonica* treated group. SEM: standard error of measurement.

**Table 5 animals-15-03517-t005:** Effects of dietary SL extract powder on sow milk composition.

	CON	SL	SEM	*p*-Value
Replicates	10	10		
Colostrum, %				
Fat	3.80	3.95	0.17	0.529
Protein	11.77	12.07	0.19	0.275
Lactose	8.58	8.47	0.18	0.653
Solids-not-fat	23.18	22.96	0.30	0.610
Milk, %				
Fat	5.79	5.33	0.42	0.446
Protein	6.46	6.40	0.08	0.629
Lactose	4.54	4.50	0.06	0.586
Solids-not-fat	11.87	11.84	0.15	0.918

CON: Control group. SL: *Scutellaria baicalensis* and *Lonicera japonica* treated group. SEM: standard error of measurement.

**Table 6 animals-15-03517-t006:** Effects of dietary SL extract on lactation performance of sows.

	CON	SL	SEM	*p*-Value
Replicates	49	48		
Litter weight, Adjusted	15.56	15.97	0.33	0.384
Litter number, Adjusted	9.60	9.63	0.19	0.938
Birth weight, Adjusted	1.63	1.66	0.02	0.223
Litter weight, Weaning	45.82	48.12	1.61	0.315
Weight, Weaning	5.46	5.51	0.12	0.800
Number Weaned	8.42	8.73	0.24	0.364
Average daily gain	202.09	202.38	6.18	0.973
Mortality	13.11% ^b^	9.72% ^a^	0.05	0.047

One sow died in the control group during pregnancy, and two sows aborted in the experimental group. Different lowercase letters in superscripts in the same line indicate significant differences (*p* < 0.05), while no letters indicate no significant differences (*p* > 0.05). CON: Control group. SL: *Scutellaria baicalensis* and *Lonicera japonica* treated group. SEM: standard error of measurement.

**Table 7 animals-15-03517-t007:** Effects of dietary SL extract powder on backfat loss and feed intake of lactating sows.

	CON	SL	SEM	*p*-Value
Replicates	49	48		
Backfat thickness, day 0, gestation, mm	21.39	21.08	0.39	0.584
Backfat thickness, day 21, weaned, mm	18.67	18.13	0.38	0.306
Backfat loss, mm	2.90	2.90	0.17	0.993
Average daily intake, kg	4.56 ^b^	4.70 ^a^	0.36	0.009
Average estrus interval, d	4.07	4.00	0.11	0.672

One sow died in the control group during pregnancy, and two sows aborted in the experimental group. Different lowercase letters in superscripts in the same line indicate significant differences (*p* < 0.05), while no letters indicate no significant differences (*p* > 0.05). CON: Control group. SL: *Scutellaria baicalensis* and *Lonicera japonica* treated group. SEM: standard error of measurement.

**Table 8 animals-15-03517-t008:** Effects of dietary SL extract powder on plasma-related hormones in sows.

	CON	SL	SEM	*p*-Value
Replicates	10	10		
Corticosterone, ng/mL				
G85	158.36	162.34	4.34	0.534
L0	142.88	146.45	4.93	0.615
L14	193.44	190.26	4.15	0.595
L21	203.24	202.07	4.64	0.861
Insulin, mIU/L				
G85	31.25	32.15	0.98	0.524
L0	27.81	28.92	0.83	0.359
L14	22.79	23.65	0.58	0.305
L21	17.86	18.47	0.49	0.391
Glucagon, ng/L				
G85	1268.18	1282.19	26.94	0.717
L0	1174.05	1148.32	21.78	0.415
L14	1316.52	1339.20	31.83	0.622
L21	1229.05	1208.29	25.02	0.565
T_3_, ng/mL				
G85	2.59	2.78	0.11	0.276
L0	3.12	3.28	0.10	0.294
L14	2.19	2.39	0.09	0.117
L21	2.63	2.74	0.10	0.498
T_4_, ng/mL				
G85	38.41	40.65	1.41	0.278
L0	41.80	43.61	1.20	0.303
L14	44.10	46.54	1.40	0.235
L21	40.05	42.66	1.40	0.205
Leptin, ng/mL				
G85	6.80	7.27	0.33	0.321
L0	6.34	6.68	0.25	0.348
L14	5.53	5.85	0.24	0.360
L21	4.69	4.78	0.19	0.748
Prolactin, ng/mL				
G85	78.54	80.92	1.63	0.314
L0	108.28	117.11	3.22	0.068
L14	100.29	106.29	2.20	0.071
L21	85.89	87.80	1.45	0.369

CON: Control group. SL: *Scutellaria baicalensis* and *Lonicera japonica* treated group. SEM: standard error of measurement. G85: Day 85 of gestation. L0: Day of farrowing. L14: Day 14 of lactation. L21: Day 21 of lactation. T_3_: Triiodothyronine. T_4_: Thyroxine.

**Table 9 animals-15-03517-t009:** Effects of dietary SL extract powder on endotoxin and HSP-70 in sows.

	CON	SL	SEM	*p*-Value
Replicates	10	10		
Endotoxin, pg/mL				
G85	488.55	474.00	8.70	0.253
L0	499.19	494.91	6.86	0.667
L14	527.93	504.70	8.48	0.069
L21	500.41	471.92	10.18	0.066
HSP-70, ng/L				
G85	2.90	2.84	0.09	0.669
L0	3.41	3.19	0.09	0.104
L14	3.70	3.42	0.10	0.082
L21	2.63	2.52	0.09	0.387

CON: Control group. SL: *Scutellaria baicalensis* and *Lonicera japonica* treated group. SEM: standard error of measurement. G85: Day 85 of gestation. L0: Day of farrowing. L14: Day 14 of lactation. L21: Day 21 of lactation. HSP-70: Heat shock protein 70.

**Table 10 animals-15-03517-t010:** Effects of dietary SL extract powder on plasma antioxidant indicators in sows.

	CON	SL	SEM	*p*-Value
Replicates	10	10		
T-AOC, mmol/L				
G85	0.56	0.57	0.02	0.661
L0	0.55	0.61	0.02	0.083
L14	0.53 ^b^	0.60 ^a^	0.10	0.024
L21	0.55 ^b^	0.60 ^a^	0.02	0.042
SOD, U/mL				
G85	69.02	70.70	10.71	0.913
L0	57.08	50.85	7.32	0.564
L14	41.73	57.07	7.97	0.212
L21	71.92	86.27	10.61	0.354
GSH-Px, U/mL				
G85	561.63	551.62	21.23	0.743
L0	647.10	660.19	20.07	0.650
L14	708.71	724.88	24.37	0.645
L21	674.57	684.58	19.31	0.718
GSH, μmol/L				
G85	7.35	7.68	1.31	0.863
L0	8.21	9.30	1.17	0.525
L14	5.41	6.81	1.04	0.351
L21	4.22	4.86	0.80	0.571
MDA, nmol/L				
G85	2.36	2.23	0.18	0.632
L0	3.01 ^a^	2.71 ^b^	0.09	0.042
L14	3.49 ^a^	3.14 ^b^	0.10	0.023
L21	4.12 ^a^	3.61 ^b^	0.15	0.028

Different lowercase letters in superscripts in the same line indicate significant differences (*p* < 0.05), while no letters indicate no significant differences (*p* > 0.05). CON: Control group. SL: *Scutellaria baicalensis* and *Lonicera japonica* treated group. SEM: standard error of measurement. G85: Day 85 of gestation. L0: Day of farrowing. L14: Day 14 of lactation. L21: Day 21 of lactation. T-AOC: Total Antioxidant Capacity. SOD: Superoxide Dismutase. GSH-Px: Glutathione Peroxidase. GSH: Glutathione. MDA: malondialdehyde.

**Table 11 animals-15-03517-t011:** Effects of dietary SL extract powder on milk antioxidant indicators in sows.

	CON	SL	SEM	*p*-Value
Replicates	10	10		
T-AOC, mmol/L				
Colostrum	1.67 ^b^	1.77 ^a^	0.03	0.039
Milk, day 14	1.32 ^b^	1.48 ^a^	0.05	0.035
SOD, U/mL				
Colostrum	70.92	71.38	11.08	0.977
Milk, day 14	50.54	55.04	6.45	0.636
GSH-Px, U/mL				
Colostrum	194.57	204.71	9.35	0.460
Milk, day 14	151.96	161.58	8.33	0.427
GSH, μmol/L				
Colostrum	5.19	5.62	0.66	0.655
Milk, day 14	4.51	4.94	0.64	0.647
MDA, nmol/L				
Colostrum	3.39 ^a^	2.61 ^b^	0.21	0.019
Milk, day 14	4.12	3.46	0.15	0.076

Different lowercase letters in superscripts in the same line indicate significant differences (*p* < 0.05), while no letters indicate no significant differences (*p* > 0.05). CON: Control group. SL: *Scutellaria baicalensis* and *Lonicera japonica* treated group. SEM: standard error of measurement. T-AOC: Total Antioxidant Capacity. SOD: Superoxide Dismutase. GSH-Px: Glutathione Pe-roxidase. GSH: Glutathione. MDA: malondialdehyde.

**Table 12 animals-15-03517-t012:** Effects of dietary SL extract powder on plasma antioxidant indicators in piglets.

	CON	SL	SEM	*p*-Value
Replicates	6	6		
T-AOC, mmol/L				
L0	0.50	0.63	0.04	0.053
L21	0.54 ^b^	0.64 ^a^	0.03	0.036
SOD, U/mL				
L0	53.90	77.65	11.22	0.200
L21	78.93	78.50	11.00	0.979
GSH-Px, U/mL				
L0	264.52	276.99	21.04	0.685
L21	288.60	309.68	29.57	0.626
GSH, μmol/L				
L0	5.04	5.78	0.71	0.491
L21	3.70	4.30	0.52	0.441
MDA, nmol/L				
L0	11.37 ^a^	10.36 ^b^	0.29	0.036
L21	6.35	5.54	0.28	0.075

Different lowercase letters in superscripts in the same line indicate significant differences (*p* < 0.05), while no letters indicate no significant differences (*p* > 0.05). CON: Control group. SL: *Scutellaria baicalensis* and *Lonicera japonica* treated group. SEM: standard error of measurement. L0: Day of farrowing. L21: Day 21 of lactation. T-AOC: Total Antioxidant Capacity. SOD: Superoxide Dismutase. GSH-Px: Glutathione Peroxidase. GSH: Glutathione. MDA: malondialdehyde.

**Table 13 animals-15-03517-t013:** Effects of a dietary SL extract powder on plasma immunoglobulin levels in serum of sows.

	CON	SL	SEM	*p*-Value
Replicates	10	10		
IgA, μg/mL				
G85	461.71	437.71	22.38	0.473
L0	530.27	555.38	22.41	0.448
L14	620.68	680.21	23.65	0.098
L21	738.09	789.81	26.74	0.193
IgG, μg/mL				
G85	1403.67	1445.51	60.84	0.634
L0	1246.99	1297.71	43.68	0.423
L14	1148.71	1251.25	38.31	0.075
L21	1054.29	1128.76	32.02	0.120
IgM, μg/mL				
G85	67.85	68.83	1.40	0.630
L0	148.68	155.36	5.34	0.399
L14	220.23	240.51	7.13	0.060
L21	341.06	358.79	14.51	0.400

CON: Control group. SL: *Scutellaria baicalensis* and *Lonicera japonica* treated group. SEM: standard error of measurement. G85: Day 85 of gestation. L0: Day of farrowing. L14: Day 14 of lactation. L21: Day 21 of lactation.

**Table 14 animals-15-03517-t014:** Effects of dietary SL extract powder on plasma immunoglobulin levels in milk of sows.

	CON	SL	SEM	*p*-Value
Replicates	10	10		
IgA, mg/mL				
Colostrum	3.74	3.88	0.15	0.039
Milk, day 14	0.54	0.60	0.03	0.035
IgG, mg/mL				
Colostrum	20.9	24.52	2.09	0.977
Milk, day 14	1.40	1.52	0.06	0.636
IgM, mg/mL				
Colostrum	1.16	1.30	0.05	0.460
Milk, day 14	0.178	0.189	0.004	0.427

CON: Control group. SL: *Scutellaria baicalensis* and *Lonicera japonica* treated group. SEM: standard error of measurement.

**Table 15 animals-15-03517-t015:** Effects of dietary SL extract powder on plasma inflammatory cytokines in sows.

	CON	SL	SEM	*p*-Value
Replicates	10	10		
TNF-α, pg/mL				
G85	104.11	102.15	2.72	0.620
L0	110.74 ^a^	104.42 ^b^	1.81	0.028
L14	110.18	101.23	3.29	0.071
L21	109.15	107.10	2.32	0.549
IL-1β, ng/L				
G85	58.04	56.26	2.51	0.623
L0	55.38	56.32	1.82	0.722
L14	58.29 ^a^	53.46 ^b^	1.33	0.020
L21	55.11	51.83	1.62	0.171
sCD14, μg/L				
G85	11.77	11.95	0.31	0.695
L0	11.18	11.79	0.33	0.207
L14	10.59 ^b^	11.68 ^a^	0.32	0.036
L21	10.96	10.89	0.19	0.810
IL-6, ng/L				
G85	235.05	229.25	6.04	0.526
L0	257.17 ^a^	236.20 ^b^	6.39	0.033
L14	245.14	230.15	5.25	0.059
L21	238.50	229.31	5.04	0.215
IL-8, ng/L				
G85	551.68	535.29	16.27	0.486
L0	592.59	571.14	13.01	0.260
L14	561.95	536.82	10.52	0.112
L21	595.35	553.83	13.78	0.050
IL-18, ng/L				
G85	218.75	216.38	5.15	0.749
L0	208.28	191.16	6.68	0.087
L14	209.33	194.89	5.64	0.092
L21	230.36	223.81	7.40	0.540

Different lowercase letters in superscripts in the same line indicate significant differences (*p* < 0.05), while no letters indicate no significant differences (*p* > 0.05). CON: Control group. SL: *Scutellaria baicalensis* and *Lonicera japonica* treated group. SEM: standard error of measurement. TNF-α: Tumor necrosis factor α. IL: Interleukin. sCD14: soluble cluster of differentiation 14.

**Table 16 animals-15-03517-t016:** Effects of dietary SL extract powder on milk inflammatory cytokines in sows.

	CON	SL	SEM	*p*-Value
Replicates	10	10		
TNF-α, pg/mL				
Colostrum	509.86	476.42	11.88	0.063
Milk, day 14	207.09	195.42	6.86	0.245
IL-1β, ng/L				
Colostrum	46.57	46.75	1.76	0.943
Milk, day 14	45.65	45.76	1.86	0.967
sCD14, μg/L				
Colostrum	7.13	7.86	0.25	0.054
Milk, day 14	7.34	7.70	0.27	0.373
IL-6, ng/L				
Colostrum	197.91	182.40	7.89	0.188
Milk, day 14	213.93	217.94	4.75	0.559
IL-8, ng/L				
Colostrum	535.58	494.35	17.22	0.109
Milk, day 14	584.17 ^a^	528.92 ^b^	11.26	0.003
IL-18, ng/L				
Colostrum	175.28	159.31	5.70	0.063
Milk, day 14	180.58	169.26	5.88	0.192

Different lowercase letters in superscripts in the same line indicate significant differences (*p* < 0.05), while no letters indicate no significant differences (*p* > 0.05). CON: Control group. SL: *Scutellaria baicalensis* and *Lonicera japonica* treated group. SEM: standard error of measurement. TNF-α: Tumor necrosis factor α. IL: Interleukin. sCD14: soluble cluster of differentiation 14.

**Table 17 animals-15-03517-t017:** Effects of dietary SL extract powder on inflammatory cytokines in piglets.

	CON	SL	SEM	*p*-Value
Replicates	6	6		
TNF-α, pg/mL				
L0	112.22	100.21	4.37	0.082
L21	95.41	87.97	3.74	0.192
IL-1β, ng/L				
L0	55.33	53.87	2.24	0.664
L21	59.81	60.04	2.01	0.939
sCD14, μg/L				
L0	8.89	9.65	0.36	0.172
L21	10.12	11.09	0.32	0.063
IL-6, ng/L				
L0	215.91	211.84	6.36	0.667
L21	236.04	230.14	7.29	0.581
IL-8, ng/L				
L0	133.84	128.03	4.30	0.384
L21	147.76	137.47	5.03	0.178
IL-18, ng/L				
L0	231.53	217.53	8.38	0.272
L21	277.71	252.29	8.26	0.062

CON: Control group. SL: *Scutellaria baicalensis* and *Lonicera japonica* treated group. SEM: standard error of measurement. L0: Day of farrowing. L21: Day 21 of lactation. TNF-α: Tumor necrosis factor α. IL: Interleukin. sCD14: soluble cluster of differentiation 14.

## Data Availability

Data are contained within the article. The datasets generated for this study are available on request from the corresponding author.

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
