# Peer review of "Dietary *Scutellaria baicalensis* and *Lonicera japonica* Extract Supplementation Attenuates Oxidative Stress and Improves Reproductive Performance in Sows"

_animals, 2025, doi:10.3390/ani15243517_

Round 1
Reviewer 1 Report
Comments and Suggestions for Authors
Add a bridging paragraph before the objectives to highlight the current research gap regarding the combined effects of Lonicera japonica and Scutellaria baicalensis in sows.
Provide the names of enzymes before using the abbreviations
Lines 150-151: Specify the equipment used for the measurements
Lines 158-159: Specify the kit number
Lines 205-212: Were the primers designed by the authors? How many samples was RNA extracted? Provide additional details on the quality assessment of RNA and cDNA.
Provide the reference for the measurement method (2^-ΔΔCt).
Lines 216-222: The authors should clarify whether data were checked for normality and variance homogeneity before analysis and specify any post-hoc tests used.
It is more formal to provide in figures Relative gene expression (mRNA) and the gene names are written in italics
Lines 439-456: Ensure consistency in reporting cytokines and immunoglobulins, and clarify whether the reductions are statistically significant across all time points or only specific days.
Lines 448-449: Rewrite the sentence (these decreases indicate is grammatically correct)
In discussion, discuss briefly the potential mechanisms linking antioxidant and anti-inflammatory effects to improved piglet survival and litter weight.
Lines 462-482: Ensure that all pathways (NF-κB, MAPK, Nrf2) and mediators (NO, ROS) are defined when first mentioned.
Lines 483-485: Rewrite the sentence for clarity
In conclusion suggest future directiosn in research field and rewrite the text for grammatical clarity
Comments on the Quality of English Language
The manuscript is well written. However, some sentences are too long and complex. A careful revision is needed to correct grammatical and syntax errors.
Author Response
Please see the attachment. Hopefully the revised version can meet your requirement. Thanks again for your help!

Reviewer 2 Report
Comments and Suggestions for Authors
Dietary Scutellaria baicalensis and Lonicera japonica Extract Supplementation Attenuates Oxidative Stress and Improves Reproductive Performance in Sows
Dear Authors,
the manuscript is interesting and quite well prepared. The positive effect of the SL extract on reducing the level of endotoxins, increasing the antioxidant potential, reducing the level of inflammatory markers and improving the reproductive performance of sows and the growth performance of offspring has been confirmed. Main part to improve in my opinion is the subsection 2.7. Statistical analysis, because in this form of manuscript the chi2 test was used, which is not applicable in case of mean values. Described comparison in case two treatments can be determined using the t-test for independent variables, and when number of treatments is higher than two one-way ANOVA must be used. Additionally, edition of text is needed: spaces between lines, the References section, …
Below I added some suggestions/comments helpful during revision process:
Line 32
p-value instead of P-value can be used (sample of 100 sows from population).
Please check to the line 484.
Lines 42-43
Small letters except of the binominal nomenclature (Scutellaria japonica) names are normally applied.
Lines 82-84
in vitro (with italics).
Please check in the entire text of manuscript.
Lines 89-360
Italics must be added to the subsection titles/headers.
Dot before title/header of subsection/sub-subsection must be added: 2.1. Experimental materials
Lines 101-104
Space between lines can be decreased.
Lines 113-114
Space between lines can be decreased.
Lines 116-118
ad libitum
Line 124
Additional information needed: L-Lysine (98-99%) or L-Lysine HCl (78%).
Lines 126-127
Space can be reduced between Table and Note below.
Lines 133-135
Space between subsection title and paragraph can be deleted.
Line 157
Zanello et al. reference required there if this method is mentioned (need to be also added to the references section).
Line 167
Power of 10 in case of density can be present in superscript form: 105/mL.
Line 175
Power of ten in the same form can be presented the same as in line 167.
Line 185
Scuttellaria baicalensis
Lines 194-196 and 205-207
Space between can be reduced.
Line 212
Table 3 with normal font without italic.
Line 216
Statistical analysis subsection must be rewritten, because the chi2 test is usually used when only numbers of observations are taking into a consideration, between observed and expected values. In this case mean values are emphasized and SEM, that is why you perhaps used the t-test for independent variables to obtain p-value. Information about parametrical test must be added to the text of subsection 2.7. Additionally, information about the normality distribution of data in treatments (Shapiro-Wilk’s test) and the homogeneity of variances between treatments (Levene’s test) can be added before information about parametrical test. Moreover, in case of Figure 1A and Figures 2-5 compared are more than two treatments, that is why one-way ANOVA analysis can be also emphasized in: 2.7. Statistical analysis subsection. Post-hoc test can be also (Tukey’s test).
Line 220
Tendency must be used instead of trends (p<0.1)
Lines 224-225
Space between lines can be reduced.
Line 231
Table 4
Last column, p-value.
Please check to Table 17.
Lines 232 and 246
Space can be reduced between Table and Note below.
Lines 236-237
Space can be reduced between lines.
Line 245
Table 6
Please check mortality (probably 100%-86.89% and 100%-90.28%).
Line 249
Table 7
Please check if superscripts 1 and 2 are required?
Lines 254-363
Indentation of first line of paragraph must be added.
Lines 266-268
One line space between can be deleted.
Line 281
Can be deleted (reduction space between paragraph and Table 10).
Line 299
Table 14?
Line 303
Table 14
p-value needed.
Lines 310-316
Tables 15, 16 and 17 must be taken into a consideration.
Line 317
Can be deleted (reduction space between paragraph and Table 15).
Lines 339-345
Title of Figure can be merged with description with the same font (one unit lower than in manuscript).
Please check to the line 387
Line 362
Can be deleted before the subsection title and the new paragraph.
Subsection title/header must be emphasized with italics.
Lines 392-501
Text must be justified.
Lines 413-483
Space between paragraphs can be deleted.
Line 496
in vivo, in vitro
‘…effect of dietary SL extract...’ has been occurred? ‘…in vivo and in vitro…’.
Lines 501-623
References section
References must be adapted to the pattern described in the Instructions for Authors.
Last name must be separate from first by comma, initial/s of name/s must have added dot, each Author must be separated by semicolon.
Journal names must be emphasized with italics, abbreviations must be used, year of publication must be emphasized with bold, volume number must be italicized
Doi number/link must be added.
I.e.:
1. Goodband, R.D.; Tokach, M.D.; Goncalves, M.A.; Woodworth, J.C.; Dritz, S.S.; DeRouchey, J.M. Nutritional enhancement during pregnancy and its effects on reproduction in swine. Anim. Front. 2013, 3(4), 68-75. https://doi.org/10.2527/af.2013-0036
Author Response
We noticed that our methods are badly described, especially "statistical analysis" part. We fully rewrite it.
Please see the attachment. Hopefully the revised version can meet your requirement. Thanks again for your help!

Reviewer 3 Report
Comments and Suggestions for Authors
Dear authors,
thank you very much for submitting your manuscript. Please finde the reviewer´s report down below.
Kind regards.
Reviewer´s report:
Summary:
In this study the authors examine antioxidant and anti-inflammatory effects of Scutellaria baicalensis and Lonicera japonica extract on sows and their offsprings during late gestation and the lactation period. They were able to show significant effects on performance parameters like litter weight and piglet mortality. Significant effects on antioxidant parameters like MDA as well as inflammatory parameters like IL-6, TNF-α and sCD14 in sows could be observed.
General comment on the hypothesis of the work:
The present study is very interesting and provides important information on antioxidant and anti-inflammatory effects of Scutellaria baicalensis and Lonicera japonica extract on sows and their offsprings.
Unfortunately, an ethical statement on the conduct of the animal experiment, including details of the authorizing authority, is missing. If this information cannot be provided, publication of the manuscript must be rejected.
The abstract and the introduction are too superficial. Detailed comments down below.
In the materials and methods section, further additions must be made to the sows, piglets, feedind, water supply, husbandry, zootechnical measurments and health status of the sows and piglets. In addition the description of materials and methods used lacks of detailed information about the manufacturers.The formatting of the tables and figures must be improved.
The discussion is detailed, but should be organized more clearly with subheadings. The paragraphs are thematically self-contained and appear to be arranged in a series. It is difficult to follow the conclusions.
In the whole manuscript the references are highlighted in grey. In addition, the reference list must be revised following the guidelines of the journal. Especially the abbreviations of the jorunals and the citation style.
According to the journal’s guidelines, the following information are missing:
- Author contribution
- Funding
- Institutional Review Board Statement (Especially the Ethical Statement for the animal experiment)
- Data Availability Statement
- Conflicts of Interest
Comments on the Abstract:
L27: Scutellaria baicalensis and Lonicera japonica (SL)
LL28-29: Please add the sizes of control and SL group.
LL30-31: day 21 of lactation
L32: Please add the mean mortality rates for the two groups.
LL32-33: Please add the mean feed intake of the sows for control and SL group.
Comments on Introduction:
L49: porcine
L53: Southern
LL58-59: Zhang et al. [8] indicated that… (The phrase “as per the information from their study in 2021” is not necessary.)
L61: Please write out the abbreviation LPS.
LL63-64: If it is possibe, please describe the mechanism of anti-inflammatory effects of Lonicera japonica in detail.
L65: Please write out the abbreviations TNF-α and IL.
LL67-71: If it is possibe, please describe the mechanism of anti-inflammatory effects of baicalin in detail.
L68: Please write out the abbreviations SOD and CAT.
L72: Please write out the abbreviations SL, T-AOC and MDA.
L74: You describe that the safety and efficacy of Scutellaria baicalensis and Lonicera japonica were confirmed in animal studies. Which standards were used to analyze the safety and efficiency? Which animal species were used in these experiments?
LL79-80: Please specify the exact stage of gestation of the sows. Day 85 was mentioned in the abstract. In addition, in the abstract you wrote in LL30-31 “until weaning day 21”. In L80 you wrote “to 21 days post-weaning of piglets”. Please specify the time period of the feeding of SL.
Comments on Material and Methods:
L88: Please remove the space between the heading and the text.
L89: Better: 2.1 Materials
L93: Please add the city, federal state and national state of the manufacturer for Sigma.
L94: Please write out the abbreviation DMEM/F12.
LL94-96: Please add the city, federal state and national state of the manufacturer for Gibco.
L96: Please add the city, federal state and national state of the manufacturer for Sigma and ScienCell.
L98: Please add the city, federal state and national state of the manufacturer for Sigma.
L101: Better: 2.2 Methods
L102: Please remove the space between the heading and the text.
L103: Animals and (space too much)
L104: Better: experiment instead of test
L105: July 1st to September 16th
L105-106: Which insemination boar was used? Please add the total number of selected sows (n = 100).
L107: What did you mean with genetics as criteria for division of the two sow groups?
L108: What did you mean with this phrase: “Each group had 50 replicants with one sow each.”?
L114: Better: according the NRC instead of meeting
L115: 2.15 x 1.15 m (missing space)
LL114-120: The description of the husbandry and feeding is too superficial and confusing. Please describe the pens, feeding system, drinking system and the health status of the sows in detail. In LL116-118 you wrote that limited feeding began on day 6 post-farrowing. In LL118-119 you wrote that the sows were fed ad libitum from days 6 – 21 of lactation. Day 6 post-farrowing and 6th day of lactation are the same. Please describe the correct feeding management.
L134: Please remove the space between the heading and the text.
LL139-140: Please describe the method of evaluation of the backfat thickness and add the manufacturer, city, federal state and national state.
L145: Why did you use the ear vein for take this large blood volume? Blood sampling from the jugular vein is faster and minimizes the risk of artifacts caused by the sampling method.
L145: Please add the manufacturer, city, federal state and national state of the manufacturer for the EDTA tubes.
L146: Please add the manufacturer, city, federal state and national state of the manufacturer for the cryotubes.
L147: Better: snap-frozen instead of frozen
L148: You take samples of 6 piglets per group. As you describe above 500 – 600 piglets were included in each group. Why did you take samples of only 1% of the piglets?
L151: Please add the city, federal state and national state of the manufacturer for Nanjing Jiacheng kit.
L152: Please add the material name, city, federal state and national state of the manufacturer for Wuhan Huamei ELISA.
L157: following Zanello et al. […]. (The phrase “method for fat removal.” Is not necessary) The citation Zanello et al. is not listed in the reference list. Please add it to the reference list and add the number to the citation in the text.
L159: Please add the material name, city, federal state and national state of the manufacturer for Nanjing Jiancheng kit and Wuhan Huamei kit.
L161: Please correct the numbering of the paragraph.
L162: Please write out the abbreviation pMEC.
L166: Please add the material name, city, federal state and national state of the manufacturer for the cultural medium.
L167: inoculate (no capital letter)
L172: Please write out the abbreviation CCK-8.
L179: Please add the city, federal state and national state of the manufacturer for the CCK-8 kit.
L180: Please add the city, federal state and national state of the manufacturer for the flow cytometer.
L185: Scutellaria baicalensis (italic style)
L195: Please remove the space between the heading and the text.
L196: Please write out the abbreviation SDS-PAGE.
L197: Please write out the abbreviation PVDF.
L198: Please write out the abbreviation TBST.
L201: Please write out the abbreviation ECL.
LL202-203: Please add the city, federal state and national state of the manufacturer for the Image software.
L206: Please remove the space between the heading and the text.
L209: Please add the city, federal state and national state of the manufacturer for the EZB kit.
L212: Why did you wrote Table 3 in italic style?
L217: Please remove the space between the heading and the text.
L218: Please add the city, federal state and national state of the manufacturer for SPSS 23.0.
Table 1:
Better entitled as: “Division of the experimental groups”
Please add the abbreviation SL to the description of the table and writ it out.
Table 2:
Proportion (%); Content (%); DE (MJ/kg); Methionine + Cysteine; Vitamin C (95%); Biotin (2%); Folic acid (98%); Choline chloride (50%) (missing space)
Please add the abbreviation DE to the description of the table and write it out.
If the table is continued on another page, please add the table header row and the information “Cont. Table 2”.
Table 3:
Please add the abbreviations bp, TNF-α and IL to the description of the table and write it out.
If the table is continued on another page, please add the table header row and the information “Cont. Table 3”.
Comments on the Results:
L233: …two sows aborted… (plural)
L238: 14th-day milk
L246: …two sows aborted… (plural)
L250: …two sows aborted… (plural)
L256: 14th day
L260: 14th and 21st days
L262: 14th day
L267: Please remove the space between the heading and the text.
L268: 10, 11 and 12 (missing space)
L274: 26.0%
LL276-277: 14th and 21st days
L279: 14th day
L296: Please write out the abbreviation Ig.
LL297, 298, 300, 307, 314: 14th day
L314: Please write out the abbreviation sCD14.
L340: levels (0, 5, 10, 25, 50, 100, 200 µg/mL) (missing space)
L349: first increased (missing f)
L350: 2µg/mL. Therefore (missing space)
L362: Please remove the space between the heading and the text.
LL366-367: Please use the correct hyphenation for the word antioxidant.
L372: expression (A) (missing space)
L372-373: inflammatory cytokines (no capital letter)
L379: Please write out the abbreviations p65, IκBα, p38, ERK, JNK.
Table 4:
P-value in the table header row.
Please add the abbreviations CON, SL, and SEM to the description of the table.
Table 5:
P-value in the table header row.
Please add the abbreviations CON, SL, and SEM to the description of the table.
Table 6:
P-value in the table header row.
Please add the abbreviations CON, SL, and SEM to the description of the table.
Table 7:
P-value in the table header row.
Please add the abbreviations CON, SL, and SEM to the description of the table.
Table 8:
P-value in the table header row.
Please add the abbreviations CON, SL, SEM, G85, L0/14/21, T3 and T4 to the description of the table.
Table 9:
P-value in the table header row.
Please add the abbreviations CON, SL, SEM, G85, L0/14/21, T3 and T4 to the description of the table.
If the table is continued on another page, please add the table header row and the information “Cont. Table 9”.
Table 10:
P-value in the table header row.
Please add the abbreviations CON, SL, SEM, G85, L0/14/21, T-AOC, SOD, GSH-Px, GSH and MDA to the description of the table.
If the table is continued on another page, please add the table header row and the information “Cont. Table 10”.
Table 11:
P-value in the table header row.
Please add the abbreviations CON, SL, SEM, T-AOC, SOD, GSH-Px, GSH and MDA to the description of the table.
Table 12:
Are the results in table 12 from sows or piglets? The sample size of 6 would suggest piglets. Please explain and adjust the table.
P-value in the table header row.
Please add the abbreviations CON, SL, SEM, L0/21, T-AOC, SOD, GSH-Px, GSH and MDA to the description of the table.
Table 13:
P-value in the table header row.
Please add the abbreviations CON, SL, SEM, G85, L0/14/21 and Ig to the description of the table.
Table 14:
Were no significance levels calculated for the results?
Please add the abbreviations CON, SL, SEM and Ig to the description of the table.
Table 15:
P-value in the table header row.
Please add the abbreviations CON, SL, SEM, G85, L0/14/21, TNF-α, IL and sCD14 to the description of the table.
Table 16:
P-value in the table header row.
Please add the abbreviations CON, SL, SEM, TNF-α, IL and sCD14 to the description of the table.
If the table is continued on another page, please add the table header row and the information “Cont. Table 16”.
Table 17:
P-value in the table header row.
Please add the abbreviations CON, SL, SEM, L0/21, IL, TNF-α and sCD14 to the description of the table.
Figure 1:
Please present the figures B-d in higher size.
Please add the abbreviations pMEC, LPS and CON to the description of the figure and write it out.
Figure 2:
Please write CON in the right figure in capital letters.
Please add the abbreviations CCK-8, CON, LPS and SL to the description of the figure and write it out.
Figure 3:
Please divide the diagrams in A = mRNA expression, and B = Inflammatory cytokines. In its current form, the division is not obvious.
Please write CON in the figures in capital letters.
Please add the abbreviations TNF-α, IL, CON, LPS and SL to the description of the figure and write it out.
Figure 4:
Please add the abbreviations TLR4, P65, p-P65, IκBn CON, SL, LPS and kDa in the description of the figure and write it out.
Figure 5:
Please add the abbreviations P38, p-P38, ERK, JNK, p-JNK, CON, SL, LPS and kDa in the description of the figure and write it out.
Comments on the discussion:
L392: Better: basis instead of foundation.
L405: baicalin (no capital letter)
L423-424: TNF-α
L438: even greater (missing space)
L442: indicate that (missing space)
L460: porcine (no capital letter)
L463: Please write out the abbreviations NO and PGE6.
L466: Please write out the abbreviation BV-2.1
L468: Glaesserella parasuis (italic style, species only in lowercase letters)
L471: Please write out the abbreviation HGF-1.
L473: Please write out the abbreviation COX-2.
L474: Please write out the abbreviation iNOS.
L476: First use of the abbreviation NO in L463.
L477: Please write out the abbreviation Nrf2.
L478: Please write out the abbreviation ROS.
Figure 6:
The figure is not referenced in the text.
The size of the figure is too small.
Comments on the Quality of English LanguageThe English language of the manuscript is generally fine, but there are some unusual words and awkward phrases that stand out. It would be beneficial to have a native English speaker review it.
Author Response

(The authors gave the same response as above.)

Round 2
Reviewer 1 Report
Comments and Suggestions for Authors
The authors have provided detailed responses to my comments and have incorporated the suggested revisions into the manuscript. Only the responses to comments 9 and 12 are currently missing.
Lines 535-536: The word human is repeated twice
Author Response
Dear Reviewer 1,
Thank you for your comment!
We apology for the revision on comment 9 and 12. Now we rewrite those 2 sentences again to meet your requirement. We highlight it in teal - I think it will be easy to find.
Also, we deleted the "human". Thank you for your attention.
Hoping hearing from you, thank you for helping us reviewing this manuscript!
Reviewer 2 Report
Comments and Suggestions for Authors
Dear Authors,
Thank you for The Revision process.
I don’t have more comments and suggestions in case of present version of manuscript.
Best regards,
Author Response
Dear Reviewer 2,
Thanks again for reviewing our manuscript. Your suggestions on statistical analysis part helped us a lot.
Hoping hearing from you.
Reviewer 3 Report
Comments and Suggestions for Authors
Dear authors,
thank you very much for incorporating my comments. The manuscript is now better structured, and the materials and methods used are comprehensible. The discussion is also clearer and more coherent. I have only a few further comments.
Kind regards.
Further comments:
L63: lipopolysaccharide (LPS) (missing space)
L66: Tumor necrosis factor α (TNF-α) (missing space)
L67: Interleukin 18 (IL-18) (missing space)
LL70-71: Superoxide Dismutase (SOD) and Catalase (CAT) (missing space)
L74: The abbreviation for Interleukin is written out in L67. Another explanation is not necessary.
LL75-76: Total antioxidant Capacity (T-AOC); malonidialdehyde (MDA) (missing space)
LL77-78: This sentence is incomplete. Please check the formulation.
L112: pregnancy (n=100) (missing space)
L157-162: Please check the space between the terms and the brackets. E.g. IL-1ß (CSB-E06782p)
L190: cytometer (Mindray...) (missing space)
L205: electrophoresis (SDS-PAGE) (missing space)
L206: difluoride (PVDF) (missing space)
L208: detergent (TBST) (missing space)
L211: chemiluminescence (ECL) (missing space)
L216: kit (EZbioscience...) (missing space)
L226: method [21] (missing space)
L232: 23.0 (IBM...) (missing space)
L236: ANOVA. Duncan's... (missing space)
L238: software (La Jolla...) (missing space)
L336: Ig (immunoglobulin) A (missing space)
L422: What does this line presents?
L432: p65 (p65); alpha (IκBα) (missing space)
L433: kinase (p38); kinase (ERK) (missing space)
L527: Oxide (NO) (missing space)
L528: E2 (PGE2) (missing space)
L538: Cyclooxygenase-2 (COX-2) (missing space)
L539: synthase (iNOS) (missing space)
L542: factor 2 (Nrf2) (missing space)
L544: species (ROS) (missing space)
LL581-587: Please add your potential conflicts of interest and delete the misplaced text.
L117, 316, 323, 330, 342, 346, 363, 369, 374, 410-411, 443, 451,: SL: Scutellaria baicalensis and Lonicera japonica (italic style)
Table 2 and Table 4: Please check the formatting of the tables before the publication of the manuscript.
Table 3: Please check the formatting of 176[R31]
In the discussion, please seperate the paragraphs with subtitles.
Author Response
Dear Reviewer 3,
Thank you for your comments!
We added all the missing spaces that you have mentioned.
For LL77-78, we have completed the sentence and highlighted it in teal.
For L422, this line was the inserted split line - now we resized the graph, combined the split line with the graph and this line disappears. Sorry for the confusion.
We discussed about the potential conflicts of interest and added it in the text.
Also, the format problem of tables was fixed.
Thank you again for the comments and it really helped us improve a lot.
Hoping hearing from you!